# RIP3 Translocation into Mitochondria Promotes Mitofilin Degradation to Increase Inflammation and Kidney Injury after Renal Ischemia–Reperfusion

**DOI:** 10.3390/cells11121894

**Published:** 2022-06-11

**Authors:** Yansheng Feng, Abdulhafiz Imam Aliagan, Nathalie Tombo, Derrick Draeger, Jean C. Bopassa

**Affiliations:** Department of Cellular and Integrative Physiology, School of Medicine, University of Texas Health Science Center at San Antonio, San Antonio, TX 78229, USA; fengy@uthscsa.edu (Y.F.); danaliagan@gmail.com (A.I.A.); ntombo22@email.mmc.edu (N.T.); draegerd@livemail.uthscsa.edu (D.D.)

**Keywords:** inner mitochondrial membrane protein (*immt*; mitofilin), acute kidney injury (AKI), mitochondrial dysfunction, receptor-interacting protein kinase 3 (RIP3), mitochondrial structural integrity and function, mtDNA release, cGAS–STING–p65 pathway, inflammation

## Abstract

The receptor-interacting protein kinase 3 (RIP3) has been reported to regulate programmed necrosis–necroptosis forms of cell death with important functions in inflammation. We investigated whether RIP3 translocates into mitochondria in response to renal ischemia–reperfusion (I/R) to interact with inner mitochondrial protein (Mitofilin) and promote mtDNA release into the cytosol. We found that release of mtDNA activates the cGAS–STING pathway, leading to increased nuclear transcription of pro-inflammatory markers that exacerbate renal I/R injury. Monolateral C57/6N and RIP3^−/−^ mice kidneys were subjected to 60 min of ischemia followed by either 12, 24, or 48 h of reperfusion. In WT mice, we found that renal I/R injury increased RIP3 levels, as well as its translocation into mitochondria. We observed that RIP3 interacts with Mitofilin, likely promoting its degradation, resulting in increased mitochondria damage and mtDNA release, activation of the cGAS–STING–p65 pathway, and increased transcription of pro-inflammatory markers. All of these effects observed in WT mice were decreased in RIP3^−/−^ mice. In HK-2, RIP3 overexpression or Mitofilin knockdown increased cell death by activating the cGAS–STING–p65 pathway. Together, this study point to an important role of the RIP3–Mitofilin axis in the initiation and development of renal I/R injury.

## 1. Introduction

Receptor-interacting protein kinase 3 (RIP3) is a member of the receptor-interacting protein (RIP) family of serine–threonine protein kinases. RIP3 is an important regulator of both programmed necrosis–necroptosis, an inflammatory form of cell death observed in pathogen-induced and sterile inflammation [1], and TNFα-induced apoptosis [2]. In the induction of necroptosis, RIP3 is a major component of the tumor necrosis factor (TNF) receptor-I signaling complex, which through additional interactions with receptor-interacting protein kinase 1 (RIP1) and pseudokinase mixed lineage kinase domain-like protein (MLKL) forms the necrosome [3,4]. Mechanistically, the formation of the necrosome requires RIP3 and MLKL but not RIPK1, shuttling into the nucleus from the cytoplasm [3] and RIP3-induced phosphorylation of MLKL [5,6]. However, the downstream sequence of intracellular events that lead to cell rupture after necrosome formation are currently unclear. Importantly, targeting the development of the necrosome could represent a novel therapeutic treatment for ischemic injury. For example, after cerebral ischemia–reperfusion (I/R), necrostatin-1 (an inhibitor of RIP1 [7]) increased the survival rate of hippocampal neurons and attenuated the impairment of learning and memory. However, RIP3 has been reported to trigger myocardial cell death through the activation of CaMKII rather than through the well-established RIP3 partners RIP1 and MLKL [8]. More recently, RIP3 activity was found to promote sepsis-induced acute kidney injury (AKI) via mitochondrial dysfunction [9]. Since I/R injury triggers numerous pathological changes, including apoptosis, oxidative stress, and inflammation [10,11,12], there is reason to believe that mitochondrial function plays a role in RIP3-dependent I/R injury.

Mitochondria play crucial roles in cell death in response to I/R stimuli [13]. At the onset of ischemia, the immediate consequence of insufficient cellular oxygenation and nutrient metabolites is the inhibition of the electron transport chain electron flow, impairing energy conservation and oxidative metabolism. Re-oxygenation leads to the production of reactive oxygen species (ROS) and a profound alteration in mitochondrial Ca^2+^ homeostasis triggering mitochondrial permeability transition (MPT), which is associated with the opening of the so-called mitochondrial permeability transition pore (mPTP) [14]. While mitochondrial dysregulation is a well-established trigger of cell death signaling cascades resulting in the necrotic pathogenesis of many diseases, including, but not limited to, I/R injury, only recent studies using patients and animal models implicated the regulation of mitochondrial cristae structures to these pathogenesis [15]. Since the Sesaki group found a role for RIP3 in neurodegeneration and the mitochondrial morphology in the absence of mitochondrial division [16], RIP3 likely plays a role in the regulation of the mitochondrial structure.

Mitochondrial inner membrane protein, Mitofilin, also referred to as Mic60, is a ubiquitously expressed mitochondrial inner membrane protein [17] initially characterized for its high abundance in rat embryonic cardiac tissue [18]. Mitofilin is part of the mitochondrial inner membrane organizing system (MINOS) required for proper maintenance of the inner membrane architecture [19], including the cristae morphology [20]. Recently, it has been found that Mitofilin acts to organize the cristae membrane, which in turn reduces cytochrome C release [21]. Additionally, the overexpression of Mitofilin preserves the mitochondrial membrane structure, leading to the restoration of the mitochondrial function [22,23,24]. Importantly, RIP3 also likely plays a role in the regulation of the mitochondrial structure, since the Sesaki group recently found that RIP3 was involved in neurodegeneration and its associated mitochondrial morphological dysfunction in the absence of mitochondrial division [16]. However, to our knowledge no studies to date have investigated whether Mitofilin regulates RIP3-dependent necroptosis.

While it is well established that degeneration of the mitochondrial crista structure results in mitochondrial dysfunction, the release of mitochondrial deoxyribonucleic acid (mtDNA) into the cytosol has emerged as a unique mechanism in the activation of intracellular pattern recognition receptors and the initiation of innate immune responses, including cGAS–STING, cytosol inflammasomes (AIM2 or NLRP3), and TLR9 [25,26]. The activation of each of these three pathways is known to increase inflammation [27,28]. Besides the nucleus, mitochondria are the only source of DNA within cells. Damage to or depletion of mtDNA directly impacts the respiratory chain, including limiting ATP production, enhancing oxidative stress, and promoting inflammatory responses, leading to tissue injury [29,30]. However, because of its probable bacterial origin, mtDNA is sensed as “foreign” once it is present within the larger cellular cytosolic compartment [27], and mtDNA release into the cytosol leads to increased transcription of pro-inflammatory markers including IL-6, TNF-α, and ICAM-1 in the nucleus [31]. While renal I/R injury triggers numerous pathological changes, including apoptosis, oxidative stress, and inflammation [10,11,12], and although recent studies determined a role for mtDNA release in AKI [32], whether mtDNA release is a mechanism of inflammation after renal I/R remains unknown.

The present study undertook investigations into the impact of Mitofilin downregulation on the RIP3-induced increase in renal I/R injury. We hypothesized that sustained Mitofilin degradation after renal I/R, in response to RIP3 translocation in mitochondria, causes mtDNA release into the cytosol. This release of mtDNA into the cytosol promotes inflammatory signaling through the activation of the cGAS–STING–p-p65 signal to facilitate the nuclear transcription of pro-inflammatory markers, including IL-6, TNF-α, and ICAM-1. We found that Mitofilin expression is downregulated with renal I/R injury. This Mitofilin downregulation is associated with mitochondrial structural damage and dysfunction after I/R in WT mice. However, mice lacking the RIP3 gene (RIP3 ^−/−^ mice) did not show this phenotype. We also found evidence that RIP3 translocates into the mitochondria in response to renal I/R, where it interacts with Mitofilin to promote damage of mitochondrial structures and facilitate mtDNA release into the cytosol. This release of mtDNA activates the cGAS–STING–p-p65 pathway to facilitate the nuclear transcription of pro-inflammatory markers, subsequently leading to exacerbation of the renal I/R injury.

## 2. Materials and Methods

### 2.1. Animals

Male adult mice (C57BL/6N, Jackson Labs) and RIP3^−/−^ mice 9–12 weeks old were used. All protocols followed the Guide for the Care and Use of Laboratory Animals (US Department of Health, NIH) and received UT Health Science Center at San Antonio Institutional Animal Care and Use Committee (IACUC) institutional approval. Animals were housed in the animal-specific pathogen-free facility at UTHSCSA’s main campus in cages with standard wood bedding and space for five mice. The animals had free access to food and drinking water and a 12 h shift between light and darkness. The animals were selected randomly and the data analysis was performed by a blinded investigator. The RIP3^−/−^ mice were a gift in kind from DR. Miho Iijima (Johns Hopkins University School of Medicine). The genotyping of RIP3^−/−^ mice was performed with the primers 5′-AGAAGATGCAGCAGCCTCAGCT-3′, 5′-ACGGACCCAGGCTGACTTATCTC-3′, 5′-GGCACGTGCACAGGAAATAGC-3′. PCR produces a 254 bp band (Rip3 deletion allele) or a 130 bp band (wild-type Rip3 allele).

### 2.2. Kidney Ischemia Reperfusion Injury

Mice were anesthetized with ketamine (80 mg/kg i.p.) and xylazine (8 mg/kg i.p.) and placed on a 37 °C heating pad. Using a midline abdominal incision, the left renal pedicle was exposed and clamped for 60 min with micro aneurysm clamps. Following this period, the clamp was removed and the abdomen was closed with 5-0 sutures. Sham-operated mice (*n* = 5/group) received identical surgical procedures, except that micro aneurysm clamps were not applied. The mice were returned to the home cage until experimentation. Blood samples and kidney tissues were collected for the next experiments.

### 2.3. Immunohistochemical Staining and Injury Assessment

The immunohistochemical analysis of Mitofilin was performed on kidney sections. The sections were deparaffined, rehydrated, quenched with 3% H_2_O_2_, and blocked with 10% normal serum, then incubated with Mitofilin antibody overnight at 4 °C, followed by incubation with a secondary antibody and avidin–biotin complex. The staining was developed by using 3′, 5′-diaminobenzidine and finalized with lightly counter staining with hematoxylin. The images were captured using Image System microscopes. The cell necrosis, loss of the brush border, cast formation, and tubular dilatation were defined as: 0, none; 1, ≤10%; 2, 11%–25%; 3, 26%–45%; 4, 46%–75%; 5, >76%.

### 2.4. Mitochondrial Isolation

Crude mitochondria were isolated from ischemic and sham kidneys. The tissue was minced and homogenized in isolation buffer A (sucrose 70 mM, mannitol 210 mM, EDTA 1 mM, and Tris-HCl 50 mM, pH 7.4) at 0.1 g of tissue/mL of buffer. The homogenate was centrifuged at 3000 rpm for 3 min, and the supernatant centrifuged for 10 min at 13,000 rpm. The resultant mitochondrial pellet was resuspended in isolation buffer B (sucrose 150 mM, KCl 50 mM, KH2PO4 2 mM, succinic acid 5 mM and Tris/HCl 20 mM, pH 7.4). The concentration was determined using the DC assay kit (Bio-Rad).

### 2.5. Mitochondrial ROS Measurement

Mitochondria ROS production was measured spectrofluorometrically (Hitachi F2710) at 560/590 nm (excitation/emission) in 100 µg of mitochondrial protein in a buffer containing 20 mM Tris, 250 mM sucrose, 1 mM EGTA, 1 mM EDTA, and 0.15% bovine serum albumin adjusted to pH 7.4 at 30 °C with continuous stirring. Amplex red dye (1 μM) (Thermofisher) and horseradish peroxidase (0.345 U/mL) were used to monitor H_2_O_2_ production, an analog for ROS generation. H_2_O_2_ levels were calculated using a standard curve of the H_2_O_2_ concentration and fluorescence intensity. A sodium salt of glutamate–malate (3 mM) was used to activate ETC complex I.

### 2.6. Ca^2+^-Induced Mitochondrial Permeability Transition

The mitochondrial resistance to the Ca^2+^-overload-induced formation of the mitochondrial permeability transition pore (mPTP) was measured. Isolated mitochondria (500 μg) were suspended in isolation buffer B with calcium green-5N dye (0.1 μM) (ThermoFisher: Cat# C3737). Samples were placed in a fluorescence spectrophotometer (Hitachi F2710) at 500/530 nm (excitation/emission) at 30 °C and incubated for 90 s followed by injections of CaCl_2_ (10 μmoles) pulses at 60 s intervals. The pulses induce a fluorescence peak of extra-mitochondrial Ca^2+^ + dye, which returns to baseline as the mitochondria absorb Ca^2+^. The calcium uptake reduces as more Ca^2+^ pulses are added, and eventually a large release of mitochondrial Ca^2+^ occurs, signaling the opening of the mPTP. The calcium retention capacity (CRC) was defined as the amount of Ca^2+^ required to initiate mPTP opening and was expressed in nmol of CaCl_2_ per mg of mitochondrial protein.

### 2.7. Mitochondrial Membrane Potential

The mitochondrial membrane potential (MMP) was qualitatively assessed in isolated mitochondria using JC-1 dye (Cayman, Cat# 15003) [33]. Mitochondria were isolated from mice kidneys in sham mice and after 60 min ischemia followed by 6 h reperfusion and treated with JC-1 (10 µg/mL) for 15 min at 37 °C with continuous gently shaking. At the end of the incubation, the media was removed and the mitochondria were washed twice with isolation buffer A. To image the cells, 0.5 mL of isolation buffer A was added to each well and images were taken using fluorescence microscopy. Polarized mitochondria are marked by punctate orange-red fluorescent staining. After depolarization, the orange-red punctate staining is replaced by diffuse green monomer fluorescence. The mitochondrial membrane potential was also assessed using Mitotracker Red (MTR, Thermofisher, Cat# M7512) [34]. Cells were seeded in 24-well plates and transfected with Mitofilin siRNA or RIP3 plasmid for 48 h. The Mitotracker solution was prepared in culture medium at a final concentration of 50 nM. The cells were incubated for 60 min in dye solution, and then washed with PBS three times. The fluorescence was measured using Image System microscopes, and the level of Mitotracker staining was define as an indication of MMP.

### 2.8. Cell Culture and siRNA Transfection

The HK-2 cell line was purchased from the American Type Culture Collection (ATCC# CRL-2190). Cells were cultured in keratinocyte serum-free medium (K-SFM) with bovine pituitary extract (BPE, 0.05 mg/mL), epidermal growth factor (EGF) 5 ng/mL, and 100 U/mL penicillin–streptomycin and grown in an atmosphere of 5% CO_2_ and 95% humidified air at 37 °C. The culture medium was changed every second day. Cells were used between passages 4 and 7 at 70–80% confluence. The siRNA against rat Mitofilin and scrambled siRNA were purchased from Life Technologies (siRNA1, Cat# 4392420; scrambled siRNA, Cat# 4390843). HK-2 cells, for passages 4–7 and at 70–80% confluence, were transfected with Mitofilin siRNA or scrambled siRNA using Lipofectamine 3000 (ThermoFisher Scientific, Cat# L3000015) according to the manufacturer’s instructions.

### 2.9. RIP3 and Mitofilin Plasmid Transfection

The PcDNA3.1+/C-flag RIP3 plasmid was purchased from Genscript (clone ID: OMu22869D). The mouse Mitofilin cDNA construct was cloned into the Myc-tagged pCDNA3.1 entry vector in the frame using the restriction enzyme XhoI/KpnI (New England Biolabs, Ipswich, MA, USA). The Mitofilin was constructed via PCR amplification using forward primer 5′-CCGCTCGAGGCCACCATGCTGCGGGCCTGTCAGTTA-3′ and reverse primer 5′-CGGGGTACCCTCTTGCTGCACTTGAGTGGT-3′. Transfections were performed using the Lipofectamine 3000 transfection reagent according to the manufacturer’s protocol.

### 2.10. Transmission Electron Microscopy

Kidney samples were fixed in 4% formaldehyde with 1% glutaraldehyde overnight at 4 °C before being washed and post-fixed for 2 h at room temperature in 2% osmium tetroxide. Sections were dehydrated in a graded alcohol series then embedded in Eponate 12 medium and cured for 48 h at 60 °C. Sections were then sliced, mounted, and stained with uranyl acetate and lead citrate and then viewed on a JEOL 1230 electron microscope to observe the mitochondrial cristae morphology and organization.

### 2.11. Immunoprecipitation

Frozen kidneys or cell samples were rapidly homogenized in immunoprecipitation *(*IP) protein lysis buffer containing 50 mM Tris-HCl, 150 mM NaCl, 1 mM Na_3_VO_4_, 0.5 mM NaF, 0.1% NP-40, and 0.25% Na-deoxycholate at pH 7.4 and supplemented with a Protease Inhibitor Cocktail tablet (Roche, 1 tablet/50 mL). The protein was precleared with Protein A/G Agarose Suspension and then incubated with 4 mg antibody at 4 °C in Protein A/G Agarose Suspension overnight. The agarose beads were collected via centrifugation and drained off of the supernatant. Finally, protein agarose beads were washed with IP lysis buffer three times and boiled for 5 min in 4× loading buffer.

### 2.12. Western Blot

Proteins from the kidney tissues, cell samples, or mitochondrial samples were extracted in IP lysis buffer containing a protease inhibitor cocktail. Protein extracts were subjected to centrifugation at 13,000× *g* for 10 min at 4 °C. Equal concentrations of lysed tissue and isolated mitochondria were loaded into 4–20% Tris–glycine gels (Bio-Rad) as recently described in [35]. Electrophoresis was carried out for 90 min at 100 V of constant voltage, followed by blotting onto nitrocellulose membranes (Millipore, Billerica, MA, USA) at 90 V for 80 min, then the membrane was blocked in 5% bovine serum albumin in Tris-buffered saline Tween (TBST, pH 7.6). Membranes were incubated overnight with different antibodies at 4 °C. Antibodies: Mitofilin (Proteintech, Cat# 10179-1-AP), cGAS (Cell Signaling Technology, Cat# 15102), GAPDH (Cell Signaling Technology, Cat# 2118), STING (Cell Signaling Technology: 13647), RIP3 (Novus Biologicals: NBP1-77299), VDAC1 (Santa Cruz Biotechnology, Cat# sc-390996), p65 (Cell Signaling Technology, Cat# 8242), p-p65 (Cell Signaling Technology, Cat# 3033), Myc-Tag (Cell Signaling Technology, Cat# 2272), Flag-tag (Cell Signaling Technology, Cat# 14793). After being washed, membranes were incubated for 1 h at room temperature with the corresponding fluorophore-conjugated secondary antibodies (LI-COR Biosciences: goat anti-rabbit Alexa 680, Cat# 926-32211 or goat anti-mouse IR Dye 800CW, Cat# 926-68070). Gels were imaged using an infrared fluorescence system (Odyssey Imaging System, Li-COR Biosciences).

### 2.13. Immunofluorescence Staining

For immunofluorescence staining, kidney sections were fixed with 4% paraformaldehyde for 15 min and permeabilized with 0.25% Triton X-100. After blocking in 3% BSA for 60 min, slides were incubated with the first antibody diluted in 1% BSA overnight. After washing with PBS, coverslips were incubated overnight with primary antibodies (Mitofilin, and RIP3) and with the secondary antibodies Alexa Fluor 488 Goat Anti-Rabbit (Abcam Cat# ab150077), and Alexa Fluor 647 Goat Anti-Mouse (Abcam, Cat# ab150119). Images were taken on a Zeiss Axiovert 200M inverted motorized fluorescence microscope (Carl Zeiss Microscope, Jena, Germany).

### 2.14. mtDNA Isolation

Kidneys were each divided into two aliquots of equal volume. One aliquot was resuspended in 500 μL DNA extraction buffer (100 mM Tris·HCl, pH 8.5, 5 mM EDTA, pH 8.0, 0.2% SDS, 200 mM NaCl, 100 μL/mL proteinase K) to extract the total DNA, which served as the normalization control for the total mtDNA. The second aliquot was resuspended in 500 μL buffer containing 150 mM NaCl, 50 mM HEPES (pH 7.4), and 25 μg/mL digitonin. The homogenates were incubated end-over-end for 10 min to allow for selective plasma membrane permeabilization and then centrifuged three times at 980× *g* for 3 min to pellet intact cells. The cytosolic supernatants were transferred to fresh tubes and spun at 17,000× *g* for 25 min to pellet any remaining cellular debris, yielding cytosolic preparations free of nuclear, mitochondrial, and endoplasmic reticulum contamination. DNA was then isolated from these pure cytosolic fractions using QIAquick Nucleotide Removal Columns (Qiagen: 28304). Quantitative PCR was performed on both whole-cell extracts and cytosolic fractions using mtDNA primers, and the cycle threshold (CT) values obtained for mtDNA abundance for whole-cell extracts served as normalization controls for the mtDNA values obtained from the cytosolic fractions.

### 2.15. RNA Isolation and Real-Time PCR

The RNA isolation was performed by following the instructions in the miRNeasy Mini Kit (Qiangen: 217004). After calculating the concentration of RNA, the Omniscript RT kit (Qiangen: 205111) was used to obtain the cDNA. The HotstarTaq master kit was used for the real-time PCR. The primers of PCR were as follows: TNF-α: Forward 5′- GAGAAAGTCAACCTCCTCTCTG-3′ and reverse 5′- GAAGACTCCTCCCAGGTATATG-3′. IL-6: Forward 5′- TAGTCCTTCCTACCCCAATTTCC-3′ and reverse 5′- TTGGTCCTTAGCCACTCCTTC-3′, ICAM-1: Forward 5′-GTGATGCTCAGGTATCCATCCA-3′ and reverse 5′- CACAGTTCTCAAAGCACAGCG-3′, β-Actin: Forward 5′-GTTGGTTGGAGCAAACATC-3′ and 5′-CTTATTTCATGGATACTTGGAATG-3′.

### 2.16. Statistical Analysis

Data presented in bar graphs are expressed as means, and error bars are the standard errors of the mean (±SEM) for a minimum of three independent trials (*n* ≥ 3). Comparisons were conducted using Student’s *t*-test and one-way ANOVA with post hoc Dunnett or Tukey corrections for multiple comparisons, where appropriate, using Prism 8 (Graphpad software). A difference of *p* < 0.05 was considered to be statistically significant.

## 3. Results

### 3.1. Renal I/R Injury Increases RIP3 Levels and Promotes Its Translocation in Mitochondria Where It Interacts with Mitofilin

To determine the effect of renal I/R on RIP3 expression, we performed a monolateral occlusion of WT mice kidneys for 60 min followed by 12 h reperfusion. Western blot analysis was performed in the whole-cell lysate to assess the levels of RIP3 in sham mice (non-ischemic) and after I/R. We found that the renal I/R significantly increased the levels of RIP3 as compared to the sham mice (185.6 ± 10.5% versus 100 ± 3%; Student’s *t*-test; *p* = 0.01; Figure 1A,B). To confirm this observation, we performed immunocytochemistry in kidney tissue sections in sham mice (non-ischemic) and after I/R using anti-RIP3 antibody. We observed a two-fold increase in RIP3 expression after I/R compared to sham animals (Figure 1A). Interestingly, the protein levels of mitochondrial RIP3 were increased following renal I/R concomitantly with a reduction in Mitofilin levels after 60 min ischemia, followed by 12, 24, and 48 h of reperfusion (Figure 1C,D). Therefore, we tested whether RIP3 and Mitofilin interact within mitochondria. To this aim, we co-transfected HK2 cells with myc-Mitofilin and flag-RIP3 plasmids. Using immunoprecipitation with myc-Mitofilin, we pulled down RIP3, while reverse IP with flag-RIP3 was used to pull down Mitofilin (Figure 2A). To confirm this interaction, we performed immunocytochemistry in kidney tissue sections in sham mice (non-ischemic) and after I/R using anti-RIP3 (green) and Mitofilin (red) antibodies. We found that the colocalization between both signals (orange-yellow) was increased after I/R compared to in sham animals (Figure 2B), suggesting an interaction between these two proteins in the inner mitochondrial membrane. Taken together, these data demonstrate that renal I/R injury increases RIP3 levels in the cytosol and promotes its translocation in the mitochondria, where it interacts with Mitofilin and facilitates a reduction in Mitofilin levels.

### 3.2. RIP3 Knockout Mice Display Decrease in Kidney Injury after Renal IR

Kidney I/R injury is known to occur after acute kidney ischemia. RIP3 is a death regulator of necroptosis, which is a programmed form of necrosis or inflammatory cell death. However, although the involvement of RIP3 in kidney I/R injury has been reported [36,37], the mechanism of RIP3 in the initiation and development of injury after renal I/R is not clear. Using our renal I/R injury model and RIP3^−/−^ mice (Figure 3A,B), we confirmed that both the renal injury score and serum creatine levels were increased in WT mice (2.607 ± 0.357 mg/dL) after I/R compared with the sham-operated animals (0.461 ± 0.126 mg/dL; *p* < 0.005; Figure 1C,D). In RIP3^−/−^ mice, renal tubular injury scores and serum creatine levels (1.360 ± 0.130 mg/dL; *p* < 0.005) were reduced compared to littermate WT mice after renal I/R (Figure 3C,D). These data confirm that RIP3 plays an important role in the development of renal I/R injury, as observed in previous studies [36,37].

### 3.3. RIP3 Knockdown Reduces the Mitofilin Degradation and Protects Mitochondrial Damage after Renal I/R

In WT mice, we found that the increase in RIP3 protein levels in mitochondria after renal I/R was concomitant with the reduction in Mitofilin levels. To confirm these results, we determined whether the knockout of RIP3 in mice leads to the reduction in Mitofilin loss in mitochondria. Using both Western blot analysis and immuno-organanello chemistry approaches, we did not observe any differences in the levels of Mitofilin in both sham groups. However, after kidney I/R injury, we found that Mitofilin levels were significantly decreased when compared to sham kidneys in WT mice, but in RIP3^−/−^ mice the levels of Mitofilin were much higher as compared to WT-I/R kidneys (Figure 4A,B). Moreover, we also found that and Oxphos were decreased after I/R injury in WT mice, but in RIP3^−/−^ mice the levels of Oxphos were higher than in WT mice after I/R injury (Figure 4C). Mitofilin is a critical organizer of mitochondrial cristae morphology and is indispensable for normal mitochondrial function. We, therefore, determined whether the preservation of Mitofilin loss observed in RIP3^−/−^ in response to renal I/R is associated with the protection of the mitochondrial structure and function. Using electron microscopy imaging, we observed the structure of WT and RIP3^−/−^ kidney mitochondria in both sham and post-renal I/R groups. We observed that kidney I/R injury seriously damaged the structure of the mitochondria compared to the sham group in WT mice. However, in RIP3^−/−^ mice, the damage to the mitochondria after renal I/R was much less when compared to the WT mice (Figure 5A). Reactive oxygen species (ROS) production in complex I of the electron transfer chain is the hallmark of mitochondrial damage [38]. We, therefore, measured the production of ROS in WT and RIP3^−/−^ kidney mitochondria in both sham and post-renal I/R groups to determine whether RIP3-induced mitochondrial damage after renal I/R is associated with mitochondrial dysfunction. We found that ROS production was significantly increased in the renal I/R group compared to the sham group in WT mitochondria. However, in RIP3^−/−^ mitochondria, the ROS production was reduced compared to WT mice after kidney I/R injury (166.70 ± 16.03 versus 207.61 ± 16.40 pmoles/min/mg of mito protein) (Figure 5B). These results were corroborated by measuring the mitochondrial membrane potential using JC-1 dye. The red fluorescence indicates healthy mitochondria, while the green fluorescence indicates depolarized mitochondria. Figure 5C shows the mitochondrial membrane potential (MMP) levels measured in isolated mitochondria from WT and RIP3^−/−^ mice, showing that both sham mice exhibited higher MMP levels than for mitochondria subjected to I/R. In fact, mitochondria from WT and RI3P^-/-^ mice subjected to I/R displayed a high ratio of aggregate (red) to monomer (green) florescent intensity compared to both sham groups (Figure 5C). We found that after I/R, mitochondria from WT and RI3P^−/−^ displayed a reduced red fluorescence intensity compared to the sham group. On the other hand, mitochondria from WT and RIP3^−/−^ mice after I/R displayed a much higher green fluorescence intensity, indicating a higher level of mitochondrial depolarization, than sham mitochondria. However, after I/R, the green fluorescence was reduced in mitochondria from RIP3^−/−^ mice compared to those from WT mice, suggesting that mitochondria from RIP3^−/−^ mice were less depolarized after I/R when compared to WT mitochondria. We report that the deleterious effect of RIP3 on mitochondria after I/R is not associated with the regulation of the mitochondrial permeability transition pore opening. We found that the mitochondrial calcium retention capacity required to induce the mPTP opening after I/R was unchanged in WT versus RIP3^−/−^ mitochondria (Figure 5D). Together, these data indicate that RIP3 regulation plays and important role in the mitochondria-dependent mechanism of kidney I/R injury. An increase in mitochondrial RIP3 levels in response to renal I/R injury might promote Mitofilin degradation, resulting in mitochondrial damage and dysfunction, presumably leading to cell death.

### 3.4. RIP3 Knockdown Reduces Mitochondrial DNA Release and Inflammatory Factors after Kidney IR Injury

Mitochondrial stress is known to cause a release of mitochondrial DNA (mtDNA) into the cytosol. We found that an increase in RIP3 levels in response to renal I/R injury results in mitochondrial damage and dysfunction. We, therefore, measured the level of cytosolic mtDNA after kidney I/R injury to determine whether the RIP3-induced increase in renal I/R injury mechanism involves the release of mtDNA into the cytosol, which is known to trigger the type Ι interferon (IFN) response. We found that the levels of mtDNA (mtNDA) were similar in both sham groups. However, after kidney I/R injury, the level of mtDNA release into the cytosol was significantly increased when compared to the sham group in WT mice, although in RIP3^−/−^ mice the level of mtDNA was reduced compared to WT-I/R (Figure 6A). We thereafter studied whether the release of mtDNA into the cytosol triggers a sequential mechanism that increases the transcription of pro-inflammatory markers including IL-6, TNF-α, and ICAM-1 in the nucleus. Because RIP3 knockdown reduces mtDNA release, we determined whether this effect was associated with the reduced production of pro-inflammatory markers. We found that the levels of pro-inflammatory factors, such as IL-6, ICAM-1, and TNF-α, were similar in both sham groups. However, after kidney I/R injury, the levels of these inflammatory factors were significantly increased compared to WT sham mice, although in RIP3^−/−^ mice the levels of these inflammatory factors were reduced compared to WT-I/R mice (Figure 6B–D). These data indicate that RIP3 knockdown in mice reduces mtDNA release from mitochondria. This effect is associated with a decrease in the transcription of pro-inflammatory makers after kidney I/R injury.

To determine the mechanism responsible for mtDNA release into the cytosol, we measured the mitochondrial calcium retention capacity (CRC). The CRC was defined as the amount of Ca^2+^ required to initiate the mitochondrial permeability transition pore (mPTP) opening. The mPTP opening is well known to play an important role in the mechanism of cell death after I/R [39]. We found no difference in CRC between WT and RIP3^−/−^ mouse mitochondria in the both sham group and after kidney IR injury (Figure 5D). These data indicate that the RIP3 knockdown-induced reduction in mtDNA release into the cytosol is not mediated by the opening of the mPTP.

### 3.5. RIP3 Knockout Deactivates cGAS–STING–P-p65 Pathway after Kidney I/R Injury

Release of mtDNA into the cytosol and out into the extracellular milieu can activate different pattern recognition receptors and innate immune responses, including cGAS–STING, cytosol inflammasomes (AIM2 or NLRP3), and TLR9. The activation of each of these pathways is able to increase inflammation [28]. Among these pathways, cGAS–STING–NFκb activation is the one that has been associated with mitochondria. We determined whether the RIP3 knockdown effects after renal I/R are related to the deactivation of the cGAS–STING–NFκb pathway. We found that the protein levels of cGAS, STING, and p-p65 were similar in both sham groups (Figure 7A–C). However, after kidney I/R injury, the levels of these proteins were significantly increased when compared to WT sham mice. However, in RIP3^−/−^ mice, the levels of these proteins were reduced when compared to WT-I/R mice (Figure 7A–C). These data indicate that the RIP3-induced reduction in mtDNA release into the cytosol decreases inflammation through deactivation of the cGAS–STING–p65 (NFκb) pathway after kidney I/R injury.

### 3.6. RIP3 Overexpression in HK-2 Cells Activates the cGAS–STING–p65 Pathway

We found that the RIP3 knockdown effects are associated with the deactivation of the cGAS–STING–p65 pathway after kidney IR injury. To confirm whether RIP3 actions are mediated via the cGAS–STING–p65 pathway, we transfected HK-2 cells with the RIP3 plasmid. We found that the protein levels of cGAS and STING were increased while Mitofilin expression was reduced when compared to the control vector (Figure 8A–F). Transfection with the RIP3-overexpressed plasmid decreased cell viability (Appendix A), increased cell death (Appendix A), and reduced mitochondrial membrane potential similar to Mitofilin knockdown with siRNA (Appendix A). Interestingly, in cells co-transfected with both RIP3-overexpressed and Mitofilin-overexpressed plasmids, the levels of cell viability were reduced when compared to cells transfected with the RIP3-overexpressed plasmid alone (Appendix A), suggesting that restoring the levels of Mitofilin can prevent cell death induced by RIP3 overexpression. Note that there was no difference in the level of cell viability between the Mitofilin-overexpressed plasmid and the control plasmid, which indicates that transfection with the Mitofilin-overexpressed vector does not induce deleterious effects in cells. These data indicate that RIP3-induced Mitofilin degradation activates the cGAS–STING pathway, resulting in increased cell death.

### 3.7. Mitofilin Knockdown in HK-2 Cells Activates cGAS–STING–p65 Pathway

We previously reported that Mitofilin knockdown in rat cardiomyoblasts by siRNA increases apoptosis via an AIF-PARP mechanism [38]. To determine whether the reduction in Mitofilin represents a key event in the mechanism of RIP3-induced cell death, we transfected HK-2 cells with Mitofilin siRNA. We found that Mitofilin knockdown increases the protein levels of cGAS and STING compared to scrambled siRNA (Figure 9A–D). These effects were associated with decreased mitochondrial membrane potential (Appendix A) and increased cell death. However, we noticed that Mitofilin knockdown did not change the levels of RIP3, suggesting that RIP3 acts upstream of Mitofilin reduction. Taken together, these data indicate that Mitofilin knockdown depolarizes mitochondria, leading to a series of events that activates the cGAS–STING pathway.

### 3.8. Inhibition of MLKL Does Not Affect Mitofilin Levels and cGAS–STING Pathway after Kidney IR Injury

RIP3 has been reported to interact with receptor-interacting protein kinase 1 (RIP1) and pseudokinase mixed lineage kinase domain-like protein (MLKL) to form the necrosome [3,4]. However, the downstream sequence of intracellular events that leads to cell rupture after necrosome formation are currently unclear. We, therefore, determined whether RIP3 translocate in the mitochondria to interact with Mitofilin and promote its loss, as well as to induce the mitochondrial dysfunction observed after the AKI mechanism involves the necrosome (RIP3/MLKL/RIP3). We found that in WT mice, cleaved RIP1 and MLKL levels were increased after AKI compared to the sham group (Figure 10A,B). However, the inhibition of MLKL with its inhibitor (necrosulfonamide–CAS 432531-71-0–Calbiochem) at different doses did not impact AKI-induced increases in cGAS and STING or reductions in Mitofilin levels observed after AKI compared to the sham group (Figure 10C). Together, these results indicate that the mechanism that underlies the activation of the RIP3–Mitofilin axis observed after AKI is not mediated by mixed lineage kinase domain-like (MLKL), which plays an important role in the pro-inflammatory necroptotic cell death program.

## 4. Discussion

In this paper, using WT and RIP3^−/−^ mice as well as an HK-2 cell line, we report that renal I/R increases RIP3 levels in the cytosol; that is, RIP3 is translocated into mitochondria where it interacts with and promotes Mitofilin degradation, which is responsible for mitochondrial structure damage and a subsequent increase in reactive oxygen species (ROS) production. We reveal that increased mitochondrial ROS production associated with mtDNA damage and release into the cytosol leads to increased nuclear transcription of pro-inflammatory markers via activation of the cGAS–STING–p-p65 pathway. The activation of this inflammatory pathway exacerbates renal I/R injury.

Although RIP3 has emerged as a critical player in cell death and a potential target to control a number of inflammatory diseases [40], to our knowledge this is the first examination of the role of RIP3 in mediating inflammation and ultimately tubular cell death after renal I/R injury [41,42]. Consistent with a role in I/R injury, we found that the protein levels of RIP3 were increased in WT mice kidneys after renal I/R compared to sham-operated animals (Figure 1A,B), and that overexpression of RIP3 increases cell death (Appendix A). RIP3 is at least in part required for the renal injury seen after I/R, since RIP3^−/−^ mice showed a significant reduction in renal I/R injury compared to their littermate WT mice (Figure 3C,D). RIP3 renal injury is also related to increased inflammatory signaling, since overexpression in HK-2 cells of RIP3 increased major inflammatory signaling proteins, cGAS, STING, and p-p65 (Figure 8), and RIP3^−/−^ mice showed significant reductions in expression levels of cGAS, STING, and p-p65 proteins (Figure 7A–C) after renal I/R injury. Taken together, RIP3 plays a role in I/R injury through increased inflammation and cell death.

In other injury models, RIP3 induces cell death by promoting distinct cell death pathways, including necroptosis [8,43] and apoptosis [44]. However, emerging evidence now suggests that RIP3-induced cell death involves critical signaling pathways from the mitochondria. Well known for their essential role in the biochemical processes of respiration and energy (ATP) production, mitochondria are also implicated in the induction of cellular death [45]. The proposed models of RIP3-induced cell death include interactions with proteins critical for mitochondrial membrane function, which result in increased ROS production [46,47]. Consistent with these models, the present study found that RIP3 overexpression in vitro reduced MitoTracker Red fluorescence, indicating an increase in mitochondrial dysfunction associated with a decrease in mitochondrial membrane potential (Appendix A). Renal I/R injury was also associated with increased mitochondrial ROS production, and mitochondria from RIP3^−/−^ displayed less structural damage, produced significantly less ROS (Figure 5B), and exhibited less depolarization (Figure 5C) compared to littermate WT mice after renal I/R injury. The increase in ROS production was not paralleled with a difference in mitochondrial Ca^2+^ retention capacity required to induce mPTP opening in WT versus RIP3^−/−^ mice, regardless of treatment (Figure 5D). This finding suggests that mitochondrial ROS generation in response to RIP3–Mitofilin axis activation, but not calcium overload, plays a larger role in the development of renal I/R injury. These results are consistent with those suggesting a role of mitochondrial ROS generation in the mitochondria-dependent mechanism of RIP3-induced cell death [9,46].

As with all organelles, the function of mitochondria is tightly linked with their structural integrity. As part of the MICOS complex, Mitofilin maintains the mitochondrial cristae morphology, intermembrane junctions, inner membrane architecture, and formation of contact sites to the outer membrane [48]. Our group recently discovered that Mitofilin expression is reduced after I/R injury of cardiac tissue, which disrupts the direct association between Mitofilin and Cyclophilin D in the IMM, increasing mitochondrial damage and increasing ROS production, ultimately leading to cardiomyocyte death [49]. Therefore, the present study examined whether Mitofilin was a downstream target for RIP3 after renal I/R injury. The initial studies confirmed that at 12, 24, or 48 h after the induction of I/R injury, increased RIP3 levels were associated with reductions in Mitofilin protein expression (Figure 1C). Additionally, RIP3 could be immunoprecipitated with Mitofilin (Figure 2A), and an immunostaining assay showed an increase in the co-localization of Mitofilin and RIP3 after renal I/R (Figure 2B), suggested very close interactions between these two proteins (Figure 2A). Using the HK-2 cell line, RIP3 overexpression promotes Mitofilin degradation (Figure 8A–F), while RIP3 knockout in mice induced deactivation of the cGAS–STING–P-p65 pathway (Figure 7A–C), and a reduction in inflammation (Figure 6B–D) is associated with the preservation of Mitofilin loss (Figure 4A,B) after kidney I/R injury. Mitofilin knockdown did not change RIP3 expression (Figure 9E–G), suggesting that Mitofilin is likely downstream of RIP3′s actions. Interestingly, knockdown of Mitofilin in HK-2 cells also increased the inflammatory markers cGAS, STING, and p-p65 (Figure 9A–G). To confirm the role of Mitofilin in the RIP3-induced cell death mechanism, HK2 cells were co-transfected with both RIP3-overexpressed and Mitofilin-overexpressed plasmids. We found that the level of the cell viability was reduced in cells co-transfected with these two vectors when compared to cells transfected with overexpressed RIP3 alone (Appendix A), suggesting that restoring the levels of Mitofilin can prevent cell death induced by RIP3 overexpression. Taken together, these results suggest a distinct mechanism in which an increase in RIP3′s interaction with Mitofilin promotes Mitofilin degradation, resulting in mitochondrial damage that increases the generation of mitochondrial ROS. However, our study did not determine the potential mechanisms whereby RIP3 translocating into mitochondria leads to Mitofilin degradation after renal I/R. We speculate that increases in Mitofilin in terms of ubiquination and oxidation are tangible mechanisms, as was recently found for an increase in Mitofilin ubiquitination in N27-A and for human dopamine neuronal primary cells treated with PD stressors, dopamine, or rotenone [34]. In addition, Mic60 is highly susceptible to oxidative stress [50], which is of particular relevance given that the mitochondrial environment in renal I/R produces high levels. Nevertheless, our findings are in line with those reported by Zhou et al., indicating that RIP3-required necroptosis positively regulated mitochondria-mediated apoptosis via the FUNDC1 mitophagy after cardiac I/R injury [51]. In addition, our result indicating that RIP3 translocation in mitochondria favors their dysfunction is supported by the report that RIP3 regulates TNF-induced reactive oxygen species production, and represents an energy metabolism regulator that switches TNF-induced cell death from apoptosis to necrosis [52]

Mitochondrial ROS can increase mitochondrial DNA (mtDNA) mutations and damage [53,54], and importantly here, I/R injury has been shown to increase the damage to cardiomyocyte membranes and subcellular structures, including mitochondrial DNA (mtDNA) [55]. One current hypothesis is that during stress conditions, including I/R, mtDNA is released into the cellular cytosol and circulation, where it acts as a pro-inflammatory signal and can increase inflammation-related injury, thereby making mtDNA release a potent pro-inflammatory mediator [56]. Here, we found that RIP3^−/−^ mice mitochondria release significantly lower levels of mtDNA into the cytosol after renal I/R injury compared to littermate WT (Figure 6A), suggesting that both RIP3 and Mitofilin can influence the release of mtDNA from mitochondria. This effect is also associated with reductions in pro-inflammatory markers, including IL-6, ICAM-1, and TNF-6 (Figure 6B–D). Our results are in line with previous reports indicating that the inhibition of the release of mtDNA from damaged mitochondria mediates EGCG-induced cardioprotective effects against I/R injury [56]. However, our findings now further confirm a role for mtDNA release in the reno-protective effect of RIP3 knockdown against renal I/R injury.

Necrosis can be categorized in uncontrolled and regulated types of cell death [57]. In controlled necrosis, also known as necroptosis, RIPK1, RIPK3, and MLKL form a necrosome [58] that initiates and executes the cell death. In the necroptosis mechanism, RIPK3 acts a downstream target of RIPK1 [5], and RIPK3 activated by RIPK1 kinase facilitates the phosphorylation and activation of MLKL. In addition, the necrosome components RIP1, RIP3, and MLKL have been reported to mediate necroptosis, which contributes to vinblastine-induced myocardial damage [59]. In this model, necroptosis results when the necrosome phosphorylates MLKL to induce the formation of a large MLKL polymer, which acts as a necroptosis effector [60]. We found that the levels of cleaved-RIP1, RIP3, and MLKL were increased after AKI (Figure 10A,B), suggesting the involvement of necroptosis in the mechanism of renal I/R injury. The mechanism proposed in this study is different, as the inhibition of MLKL, the terminal mediator in the necroptotic pathway, did not prevent the increase in Mitofilin degradation or the activation of the cGAS–STING pathway observed in renal I/R injury (Figure 10C). In addition, whether or not necroptosis requires mitochondria in vivo has yet to be fully determined [61]. Nevertheless, the translocation of RIP3 in mitochondria, where it interacts with the mitochondrial protein glutamate dehydrogenase 1 (GLUD1), has been reported by the Han group [52]. Here, we propose that RIP3 translocates into mitochondria in response to AKI stimuli, where it interacts with Mitofilin and promotes Mitofilin loss, which subsequently leads to mitochondrial structure damage and dysfunction. Our finding suggests that a reduction in RIP3 translocation into the mitochondria can preserve Mitofilin expression, which induces protective effects against renal I/R injury. Our finding might open the possibility of targeting the RIP3–Mitofilin axis to treat the adverse effects of renal I/R injury.

## 5. Conclusions

We demonstrated that kidney I/R increases RIP3 levels in the cytosol and that RIP3 translocates into mitochondria, where it interacts with Mitofilin and promotes its degradation, leading to increased mitochondrial structural damage and dysfunction. The subsequent increase in ROS production in the mitochondria is postulated to facilitate mtDNA damage and release into the cytosol, where the mtDNA activates the cGAS–STING–p-p65 pathway, leading to amplified nuclear transcription of pro-inflammatory markers that subsequently increase renal I/R injury (Figure 11).

## Figures and Tables

**Figure 1 cells-11-01894-f001:**
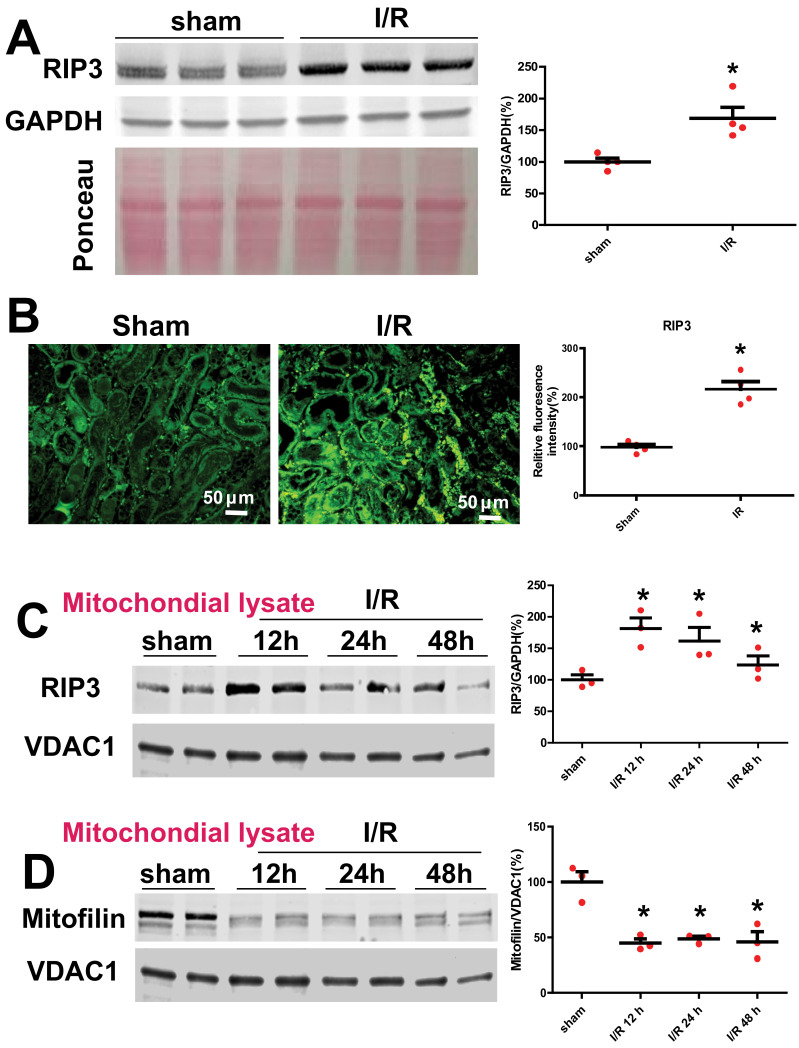
Renal I/R increases RIP3 expression and reduces Mitofilin levels. (**A**) Immunoblot and graph showing increases in RIP3 levels after 60 min of mono-lateral kidney ischemia followed by 12 h of reperfusion compared to sham animals (non-ischemic). Values are expressed as means ± SEM; * *p* < 0.05 versus sham group (*n* = 4/group). (**B**) Confocal images and graph showing increases in RIP3 levels in the kidney sections after 60 min ischemia followed by 12 min reperfusion compared to sham animals (non-ischemic). Values are expressed as means ± SEM; * *p* < 0.05 versus sham group (*n* = 4/group). (**C**) Immunoblot and graph showing increases in RIP3 levels in isolated mitochondria from kidney lysate in sham animals (non-ischemic) after 60 min of ischemia followed by several durations of reperfusion (12, 24, and 48 h) compared to sham animals (non-ischemic). This result indicates that RIP3 translocates in mitochondria after AKI. Values are expressed as means ± SEM; * *p* < 0.05 versus sham group (*n* = 4/group). (**D**) Immunoblot and graph showing reductions in Mitofilin levels in the isolated mitochondria from kidney lysate in sham animals (non-ischemic) and after 60 min of ischemia followed by several durations of reperfusion (12, 24, and 48 h) compared to sham animals (non-ischemic). Values are expressed as means ± SEM; * *p* < 0.05 versus sham group (*n* = 4/group).

**Figure 2 cells-11-01894-f002:**
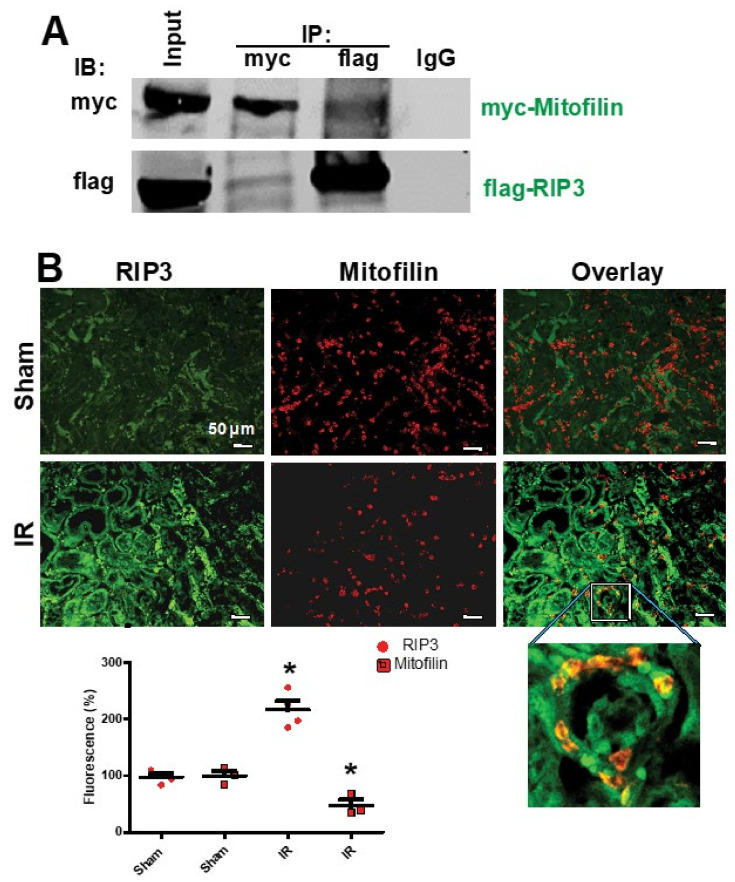
Renal I/R stress increases RIP3 interaction with Mitofilin in mitochondria. (**A**) Immunoblots showing the link between RIP3 and Mitofilin in the inner mitochondrial membrane of HK-2 cells. Immunoprecipitation (IP) in whole-cell lysate fractions with myc-Mitofilin antibody immunoprecipitated flag-RIP3, which was revealed by Western blot analysis. Reverse Co-IP with flag-RIP3 confirmed the interaction between these two proteins, as myc-Mitofilin was detected in the immunoprecipitate; *n* = 3 independent experiments. (**B**) Confocal microscopy images of kidneys from sham animals and after AKI labeled with RIP3 (green) and Mitofilin (red) and the overlay of both proteins (yellow). The graph shows the increase in RIP3 and the reduction in Mitofilin levels after AKI compared with sham kidneys. These results indicate that the co-localization between Mitofilin and RIP3 is increased after AKI (60 min ischemia followed by 12 h reperfusion); * *p* < 0.05 versus sham group, respectively, *n* = 3 experiments.

**Figure 3 cells-11-01894-f003:**
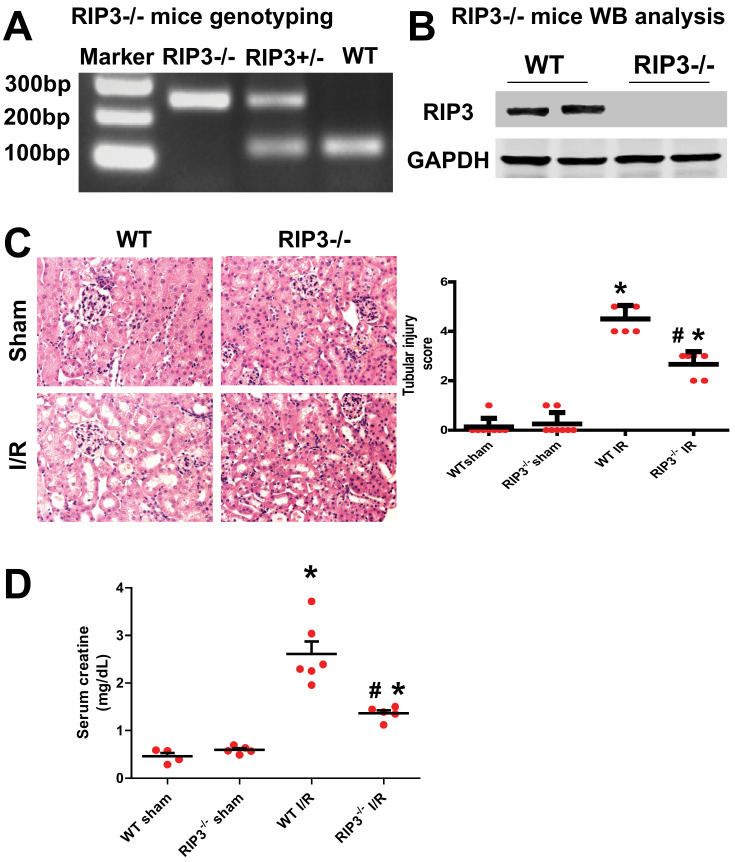
RIP3^−/−^ mice exhibit a decrease in kidney injury after renal I/R. (**A**) Genotype used to confirm the RIP3^+/−^ and RIP3^−/−^ mice versus the littermate WT mouse. Note that the RIP3^+/−^ heterozygote mouse displayed both alleles (knockout and WT bands). (**B**) Immunoblot confirming the knockdown of RIP3 in RIP3^−/−^ versus littermate WT mice. (**C**) Left: H&E staining images showing an increase in kidney injury in WT mice after renal I/R compared to the sham-operated animals. However, in RIP3^−/−^ mice, the tubular injury was reduced when compared to littermate WT mice after renal I/R. Right: Graph showing an increase in kidney injury score in WT mice after renal I/R compared to the sham-operated animals. However, in RIP3^−/−^ mice, the tubular injury was reduced when compared to littermate WT mice after renal I/R. Note that there was no difference in kidney injury between WT and RIP3^−/−^ sham mice. In addition, the cell necrosis, loss of the brush border, cast formation, and tubular dilatation were defined as: 0, none; 1, ≤10%; 2, 11%–25%; 3, 26%–45%; 4, 46%–75%; 5, >76%. Values are expressed as means ± SEM; * *p* < 0.05 versus WT group; ^#^
*p* < 0.05 versus WT-I/R group, (*n* = 6/group). (**D**) Graph showing increases in the levels of serum creatine in WT mice after renal I/R compared to sham-operated animals. However, in RIP3^−/−^ mice, the level of serum creatine was reduced when compared to littermate WT mice after renal I/R. Note that there was no difference in the levels of serum creatine between WT and RIP3^−/−^ sham mice. Values are expressed as means ± SEM; * *p* < 0.05 versus WT-Sham group, respectively; ^#^
*p* < 0.05 versus WT-I/R group (*n* = 6/group).

**Figure 4 cells-11-01894-f004:**
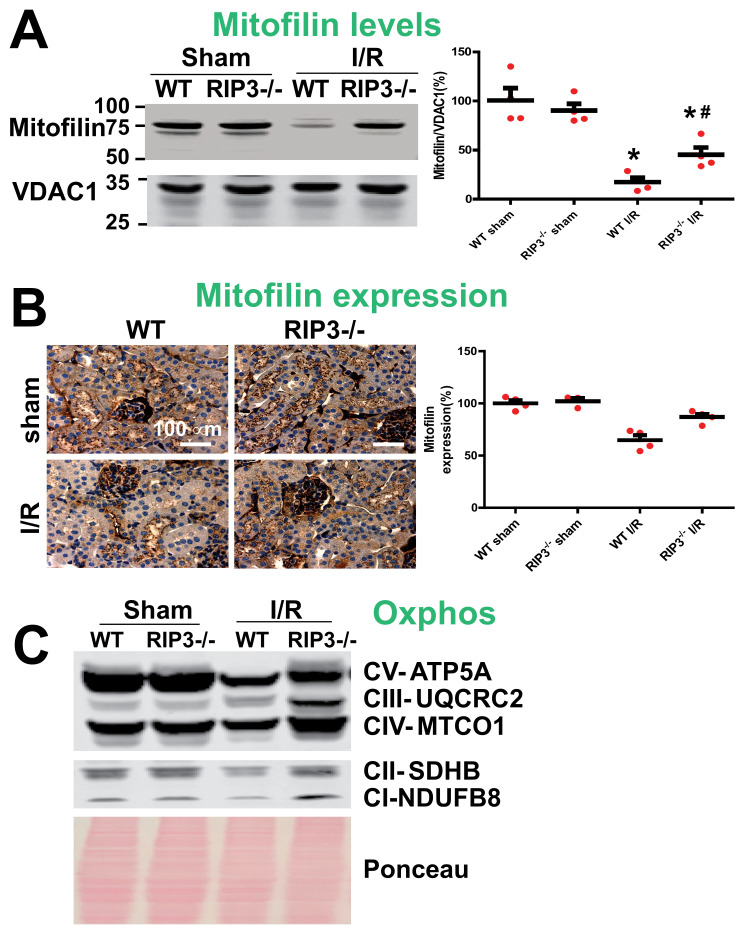
Preservation of Mitofilin in RIP3^−/−^ mice is associated with decreased mitochondrial damage and dysfunction after renal I/R. (**A**) Immunoblots showing dramatic reductions in Mitofilin levels after renal I/R compared to sham mitochondria in WT mice. However, in RIP3^−/−^ mice, the levels of Mitofilin were higher when compared to littermate WT mice after renal I/R. Note that there was no difference in the levels of Mitofilin between WT and RIP3^−/−^ sham mice. Values are expressed as means ± SEM; * *p* < 0.05 versus WT group; ^#^
*p* < 0.05 versus WT-I/R group (*n* = 4/group). (**B**) Images and graph of kidney tissues labeled with Mitofilin showing a dramatic reduction in Mitofilin signaling after renal I/R compared to sham mitochondria in WT mice. However, in RIP3^−/−^ mice, the level of Mitofilin signaling was higher when compared to littermate WT mice after renal I/R. Note that there was no difference in the levels of Mitofilin signaling between WT and RIP3^−/−^ sham mice. Values are expressed as means ± SEM; * *p* < 0.05 versus WT group; ^#^
*p* < 0.05 versus WT-I/R group (*n* = 4/group). (**C**) Immunoblots showing dramatic reductions in Oxphos (electron transport chain complex) levels after renal I/R compared to sham mitochondria in WT mice. Note the preservation of the levels of Oxphos complexes in RIP3^−/−^ mitochondria compared to WT mitochondria after renal I/R.

**Figure 5 cells-11-01894-f005:**
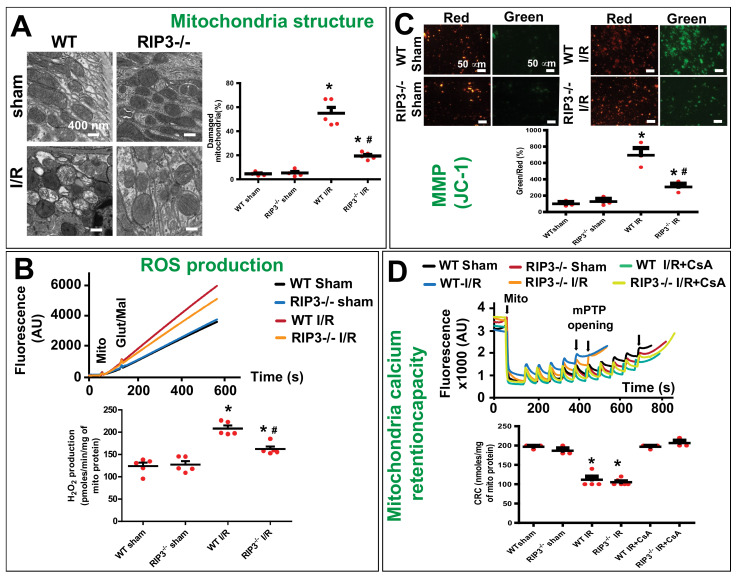
RIP3 knockout protects against mitochondrial damage and dysfunction after renal I/R. (**A**) Electron microscopy images of mitochondria in kidney tissues showing a dramatic increase in cristae disruption after renal I/R compared to sham mitochondria in WT mice. However, in RIP3^−/−^ mice, the level of cristae disruption was reduced when compared to littermate WT mice after renal I/R. Note that there was no difference in the levels of cristae disruption between WT and RIP3^−/−^ sham mice. Bar graph showing percentage of damaged mitochondria in each group. Fragmented or disrupted cristae with empty spaces (in the matrix) were considered damaged mitochondria, while mitochondria with dense continuous cristae were considered as good or undamaged. A minimum of 100 mitochondria were counted in each group. Values are expressed as means ± SEM; * *p* <0.05 versus WT sham group, ^#^
*p* <0.05 versus WT-I/R group (*n* = 5/group). (**B**) Top: Recording of mitochondrial reactive oxygen species (ROS) production using Amplex red in the presence of horseradish peroxidase after stimulation of complex I with glutamate–malate. Bottom: Graph showing a dramatic increase in ROS production after renal I/R compared to sham group in WT mice mitochondria. However, in RIP3^−/−^ mice, the ROS production was reduced when compared to littermate WT mice after renal I/R. Note that there was no difference in ROS production between WT and RIP3^−/−^ sham mice. Values are expressed as means ± SEM; * *p* <0.05 versus WT sham group, ^#^
*p* <0.05 versus WT-I/R group (*n* = 4/group). (**C**) RIP3 knockout in mice protects the mitochondrial membrane potential as measured with the JC-1 dye stained in isolated mitochondria. Image (right) and graph (left) showing an increase in the green/red fluorescence intensity ratio (that indicate depolarization) in WT mitochondria after renal I/R compared to sham mitochondria. Note the reduction in the green/red fluorescence intensity ratio in RIP3^−/−^ mice compared to WT mitochondria after renal I/R. Values are expressed as means ± SEM; * *p* <0.05 versus WT sham group, ^#^
*p* < 0.05 versus WT-I/R (*n* = 3/group). (**D**) RIP3 knockout in mice does not affect mitochondrial Ca^2+^ retention capacity (CRC) required to induce the mitochondrial permeability transition pore (mPTP) opening after renal I/R injury. Typical spectrofluorometric recordings of Ca^2+^ overload in mitochondria isolated from hearts after 60 min ischemia followed by 6 h reperfusion. Subsequently, 20 nmol-mg−1 of protein Ca^2+^ pulses was delivered until a spontaneous and massive release was observed, presumably to the opening of the mPTP (arrows). The graph shows no difference in mitochondrial CRC between WT and RIP3^−/−^ mitochondria from the sham group after renal I/R and after renal I/R supplemented with cyclosporine A (mPTP opening inhibitor, 2 µM). Values are expressed as means ± SEM; * *p* < 0.05 versus WT sham group (*n* = 3/group).

**Figure 6 cells-11-01894-f006:**
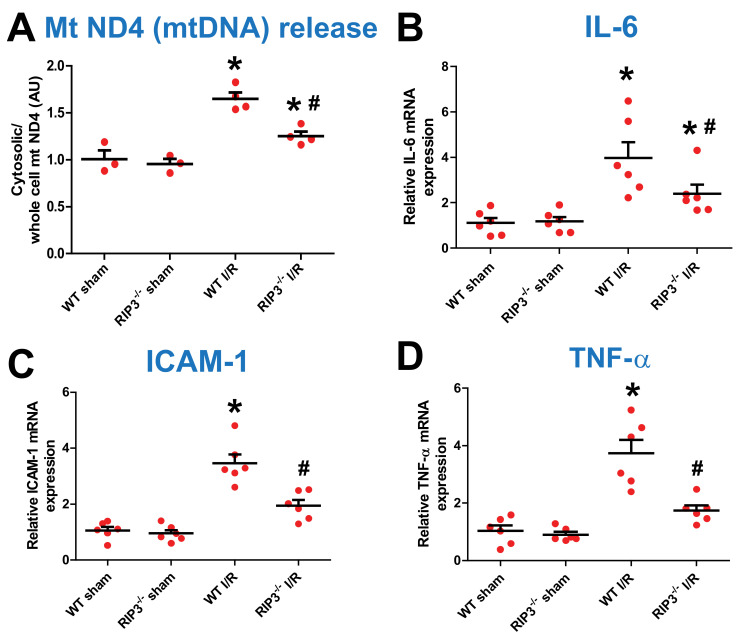
RIP3^−/−^ mice exhibit decreased mtDNA release and inflammation after renal I/R. (**A**) Graph showing an increase in mitochondrial DNA (mtDNA) release in the cytosol in WT mice after renal I/R compared with the sham-operated animals. However, in RIP3^−/−^ mice, the release of mtDNA was reduced when compared to littermate WT mice after renal I/R. Note that there was no difference in mtDNA levels between WT and RIP3^−/−^ sham mice. Values are expressed as means ± SEM; * *p* < 0.05 versus WT sham group, ^#^
*p* < 0.05 versus WT-I/R (*n* = 4/group). (**B**–**D**) Graph showing increases in pro-inflammatory markers including IL-6, ICAM-1, and TNF-6 levels in the cytosol in WT mice after renal I/R compared with the sham-operated animals. However, in RIP3^−/−^ mice, the release of these pro-inflammatory markers was reduced when compared to littermate WT mice after renal I/R. Note that there was no difference in pro-inflammatory markers between WT and RIP3^−/−^ sham mice. Values are expressed as means ± SEM; * *p* < 0.05 versus WT sham group, ^#^
*p* < 0.05 versus WT-I/R (*n* = 6/group).

**Figure 7 cells-11-01894-f007:**
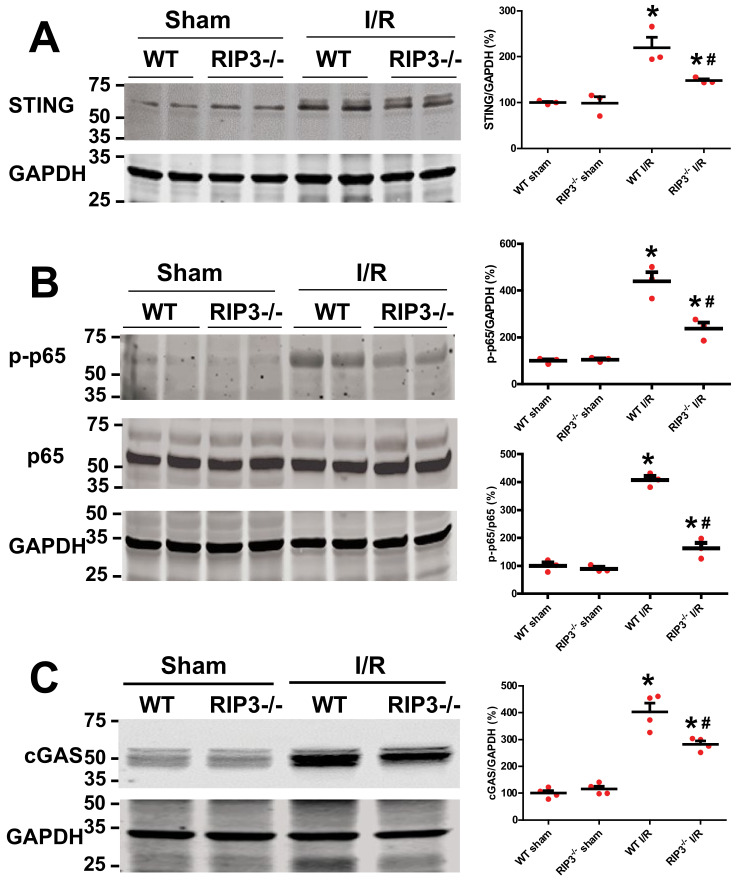
RIP3^−/−^ mice exhibit decreased levels of cGAS, STING, and p-p65 after renal I/R. (**A**) Immunoblots and graph showing increases in STING levels after renal I/R compared to sham group in WT mice. However, in RIP3^−/−^ mice, the levels of STING were reduced when compared to littermate WT mice after renal I/R. Note that there was no difference in the levels of STING between WT and RIP3^−/−^ sham mice. Values are expressed as means ± SEM; * *p* < 0.05 versus WT sham group; ^#^
*p* < 0.05 versus WT-I/R (*n* = 3/group). (**B**) Immunoblots and graph showing an increase in p-p65 levels after renal I/R compared to sham in WT mice. However, in RIP3^−/−^ mice, the levels of p-p65 were reduced when compared to littermate WT mice after renal I/R. Note that there was no difference in the levels of p-p65 between WT and RIP3^−/−^ sham mice. Values are expressed as means ± SEM; * *p* < 0.05 versus WT sham group; ^#^
*p* < 0.05 versus WT-I/R (*n* = 4/group). (**C**) Immunoblots and graph showing increases in cGAS levels after renal I/R compared to sham in WT mice. However, in RIP3^−/−^ mice, the levels of cGAS were reduced when compared to littermate WT mice after renal I/R. Note that there was no difference in the levels of cGAS between WT and RIP3^−/−^ sham mice. Values are expressed as means ± SEM; * *p* < 0.05 versus WT sham group; ^#^
*p* < 0.05 versus WT-I/R (*n* = 4/group).

**Figure 8 cells-11-01894-f008:**
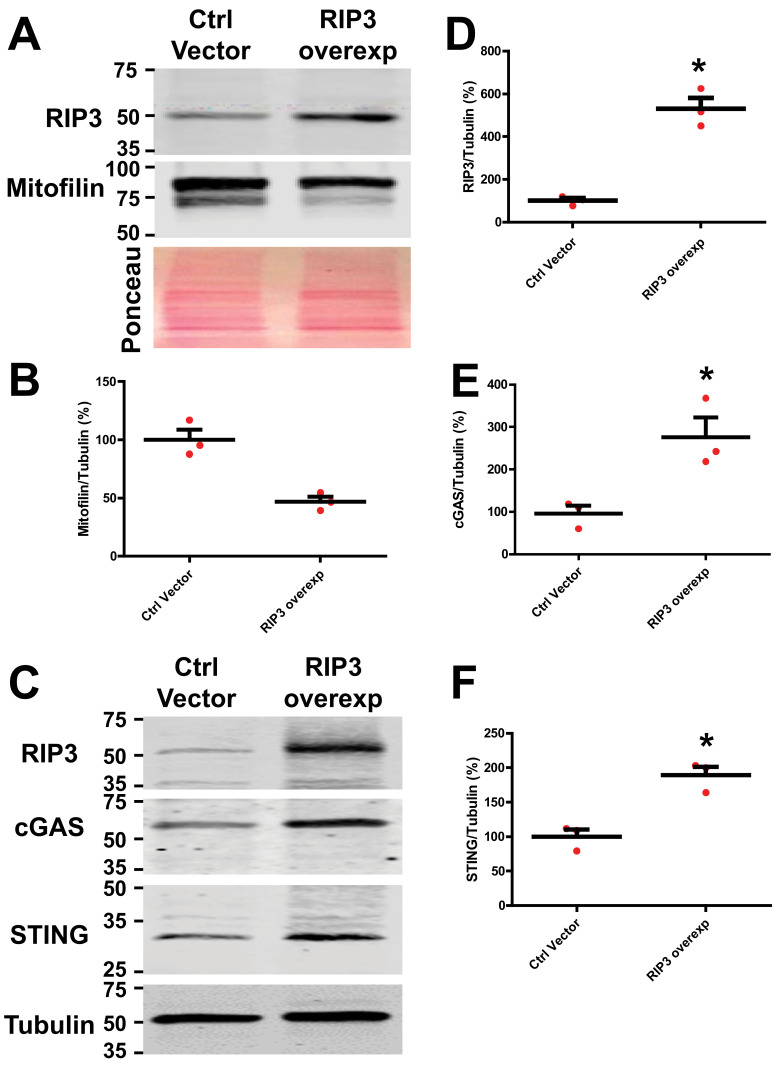
RIP3 overexpression in HK-2 cells increases the levels of cGAS and STING and reduces Mitofilin levels. (**A**) Immunoblots and graph (**B**) showing reductions in Mitofilin levels in HK-2 cells transfected with an RIP3-overexpressed plasmid compared to those transfected with pCMV6 control vector. (**C**) Immunoblots showing increases in the levels of cGAS and STING levels in HK-2 cells transfected with an RIP3-overexpressed plasmid compared to those transfected with pCMV6 control vector (**D**–**F**). Graph showing increases in the levels of RIP 3 (**D**), cGAS (**E**), and STING (**F**) in HK-2 cells transfected with an RIP3-overexpressed plasmid compared to those transfected with pCMV6 control vector. Values are expressed as means ± SEM; * *p* < 0.05 versus control (Ctrl) vector group (*n* = 3/group).

**Figure 9 cells-11-01894-f009:**
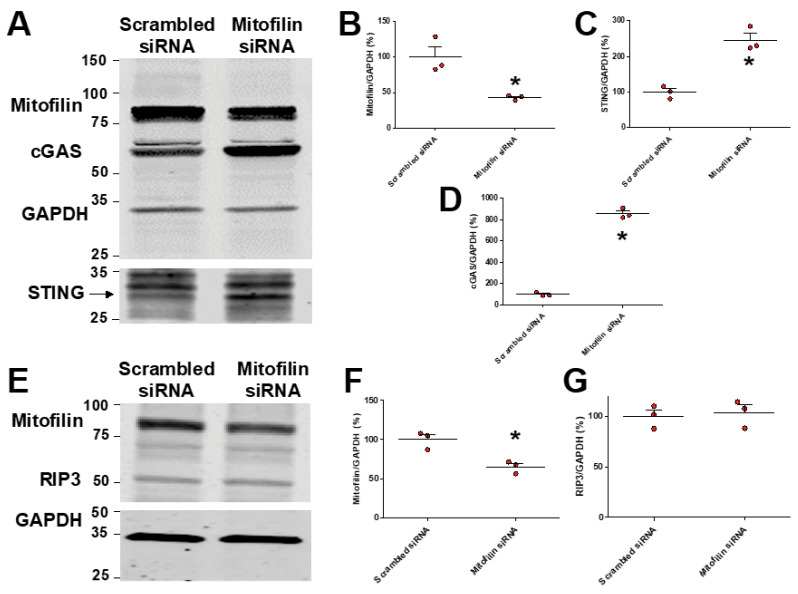
Mitofilin knockdown in HK-2 cells increases the levels of cGAS and STING, with no effect on RIP3 levels. (**A**) Immunoblots showing increases in the levels of cGAS and STING in HK-2 cells transfected with Mitofilin siRNA compared to those transfected with scrambled siRNA (*n* = 3 experiments). (**B**–**D**) Graph showing increases in the levels of cGAS (**D**) and STING (**C**) and reductions in Mitofilin (**B**) levels in HK-2 cells transfected with Mitofilin siRNA compared to those transfected with scrambled siRNA. Values are expressed as means ± SEM; * *p* < 0.05 versus control (Ctrl) vector group (*n* = 3/group). (**E**) Immunoblots showing no difference in the levels of RIP3 in HK-2 cells transfected with Mitofilin siRNA compared to those transfected with scrambled siRNA (*n* = 3 experiments). (**F**,**G**) Graph showing reductions in Mitofilin (**F**) levels and no change in RIP3 levels (**G**) in HK-2 cells transfected with Mitofilin siRNA compared to those transfected with scrambled siRNA. Values are expressed as means ± SEM; * *p* < 0.05 versus control (Ctrl) vector group (*n* = 3/group).

**Figure 10 cells-11-01894-f010:**
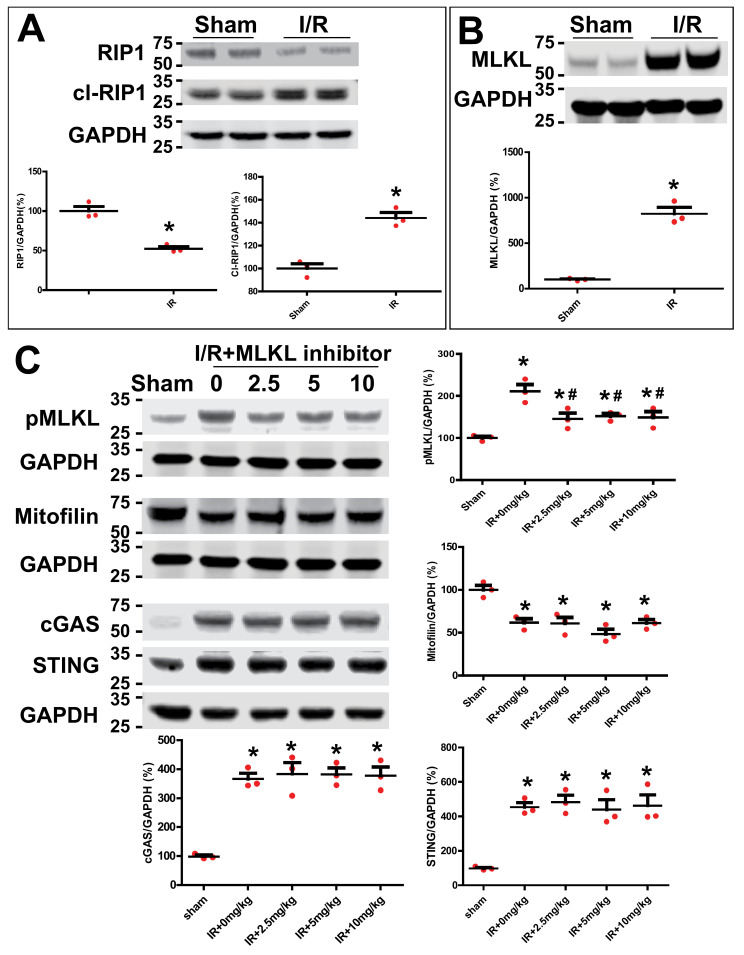
Inhibition of MLKL does not affect cGAS–STING pathway in renal I/R injury. (**A**) Immunoblot and graph showing increases in the levels of cl-RIP1 and reductions in RIP1 levels after renal I/R injury compared to sham group. Values are expressed as means ± SEM; * *p* < 0.05 versus sham group (*n* = 3/group). (**B**) Immunoblot and graph showing dramatic increases in MLKL levels after renal I/R injury compared to sham group. Values are expressed as means ± SEM; * *p* < 0.05 versus sham group (*n* = 3/group). (**C**) Immunoblots showing no changes in cGAS, STING, and Mitofilin levels in kidneys treated with different doses (0, 2.5, 5, and 10 mg/kg) of MLKL inhibitor after renal I/R injury. Note that all doses of the MLKL inhibitor were able to significantly reduce the MLKL phosphorylation; * *p* < 0.05 versus sham group; ^#^
*p* < 0.05 versus I/R+0mg/kg group (*n* = 3/group).

**Figure 11 cells-11-01894-f011:**
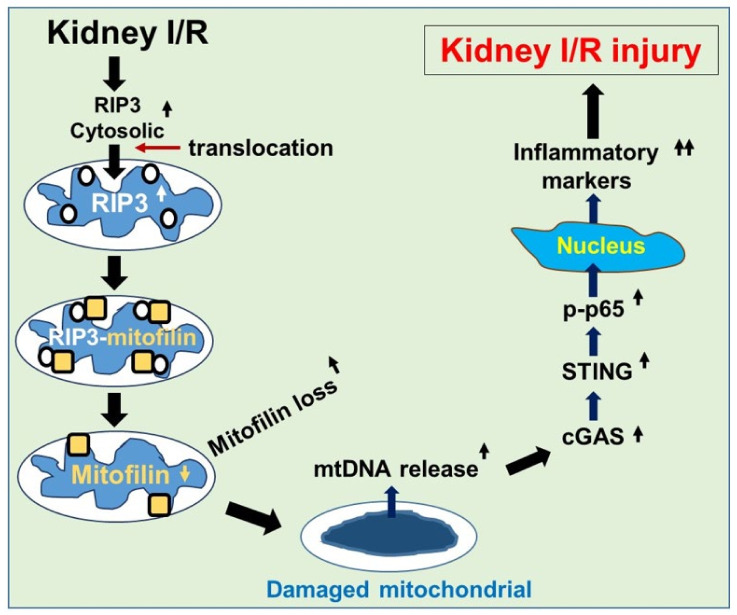
Graphic abstract summarizing the proposed mechanism whereby increased RIP3 in response to renal I/R reduces Mitofilin levels to amplify inflammation and exacerbates kidney injury. An increase in cytosolic RIP3 levels in response to kidney I/R promotes the translocation of RIP3 into the mitochondria, where it interacts with and favors Mitofilin degradation, leading to increased mitochondrial structural damage and dysfunction. The following increase in ROS production in the mitochondria is proposed to enable mtDNA damage and release into the cytosol, where the mtDNA activates the cGAS–STING–p-p65 pathway, leading to increased nuclear transcription of pro-inflammatory markers that subsequently exacerbate renal I/R injury.

## Data Availability

Not applicable.

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
