# Peer review of "RIP3 Translocation into Mitochondria Promotes Mitofilin Degradation to Increase Inflammation and Kidney Injury after Renal Ischemia–Reperfusion"

_cells, 2022, doi:10.3390/cells11121894_

Round 1

Reviewer 1 Report

Authors have established the cohesive link between RIP3 translocation and Mitofilin degradation in AKI. In addition, they have established that MLKL inhibition does not impact cGAS/STING pathway. The methodology, scientific rigor of the manuscript is good. Manuscript is nicely written and the mitochondrial biology is part is interesting. However, before acceptance of the manuscript i have some minor suggestions:

  1. 2.15 second line its concentration
  2. I strongly suggest repeating confocal microscopy for co-localization of RIP3 and Mitofilin. The images are not convincing.
  3. RIP3-/- has leaky expressions usually knockouts have absolute no expression of proteins
  4. Similarly the Mitofilin expression does not seems very obvious. It somewhere stains the brush border epithelial. How about repeating the stain with frozen section and staining the brush border epithelium with SGLT-2 to segregate it as mitochondrial protein (Confocal microscopy)
  5. Authors should check the figure legend of Figure S2 i believe cells were transfected with RIP3 not with Mitofilin.
  6. Likewise authors should read the last three lines of results 3.8.. which plays important role pro-inflammatory necroptotic cell death.
  7. Molecular weights in Figure 10.C
  8. Authors should shed some lights on clinical importance of this pathway. How it can be translated clinically.  

Author Response

Reviewer 1.

  1. 2.15 second line its concentration

We agree with the reviewer’s comment, and have updated the manuscript accordingly.

  1. I strongly suggest repeating confocal microscopy for co-localization of RIP3 and Mitofilin. The images are not convincing.

We have performed three independent experiments of AKI in both groups by immune-labelling kidney tissues with RIP3 (green) and Mitofilin (red) and the overlay of both proteins is in yellow. Several images have been taken that all show an increase in the co-localization between the two signals after I/R when compared to sham. Note that in these images the co-localization signal is underestimated because of the reduction of Mitofilin after 12 min reperfusion as shown in Fig. 1D.

  1. RIP3-/- has leaky expressions usually knockouts have absolute no expression of proteins.

We agree with the reviewer’s comment, and have reduced the exposition duration of the membrane in the Fig. 3B.

  1. Similarly the Mitofilin expression does not seems very obvious. It somewhere stains the brush border epithelial. How about repeating the stain with frozen section and staining the brush border epithelium with SGLT-2 to segregate it as mitochondrial protein (Confocal microscopy).

We have analyzed all our images and can confirm that they are conclusive. This observation is also confirmed by the Western blot analysis result (Fig. 1D). In addition, we have been given two weeks to answer to the reviewer’s comments. Because the tissues are performed in core facility that have almost 6 to 8 weeks delay. We agree with the reviewer suggestion, and would be kind to repeat this confocal microscopy images. However, the feasibility in our facility is just too long. We would appreciate that the reviewer understand that we did not refuse to repeat this study.

  1. Authors should check the figure legend of Figure S2 i believe cells were transfected with RIP3 not with Mitofilin.

We agree with the reviewer’s comment, and have edited the figure legend of the Fig. S2.

  1. Likewise authors should read the last three lines of results 3.8.. which plays important role pro-inflammatory necroptotic cell death.

We agree with the reviewer’s comment, and have edited the text accordingly.

  1. Molecular weights in Figure 10.C

We agree with the reviewer’s comment, and have added the molecular weights in Figure 10C.

  1. Authors should shed some lights on clinical importance of this pathway. How it can be translated clinically.  

We agree with the reviewer’s comment, and have added a small paragraph in the discussion to show the clinical significance of the proposed mechanism.

Reviewer 2 Report

In the manuscript entitled “RIP3 Translocation into Mitochondria Promotes Mitofilin Degradation to Increase Inflammation and Kidney Injury after Renal Ischemia/Reperfusion”, the authors revealed an important role of the RIP3/Mitofilin axis in the initiation and development of renal I/R injury. In mice model, the authors found renal ischemia/reperfusion increase RIP3 protein expression and RIP3 translocated into mitochondria. Futhermore, RIP3 interacted with Mitofilin and promote its degradation in mitochondria, which results in increased mitochondria damage and the release of mtDNA into the cytosol. Released mtDNA activited cGAS/STING/P65 pathway and increased transcription of pro-inflammatory markers which was independently on RIP1/RIP3/MLKL pathway. This paper is a topic of interest to the researchers in the related research field but it needs improvement before acceptance for publication.  

The  major concerns:

  1. RIP3 has been reported to interaction with receptor-interacting protein kinase 1(RIP1) and pseudokinase mixed lineage kinase domain-like protein (MLKL) to form the necrosome. The important roles of RIP3-MLKL-dependent necroptosis in acute kidney injury has been proved[1–3]. The author declared RIP3 translocation in mitochondria to interact with Mitofilin and promote its loss as well as mitochondria dysfunction observed after AKI is undependently on the necrosome (RIP3/MLKL/RIP3). For these, the protective effect of MLKL inhibitor (Necrosulfonamide) on acute kidney injury is very important data. The manuscript only provided Figure-10C, which is not enough to support the results. Considered necrosulfonamide has not be reported in AKI, strongly suggest to provide the details of animal study, including dosage selection, route and times of administration et.al. The key data, including renal histopathology, mtDNA, inflammatory gene and protein(IL-6, TNF-α and ICAM-1), p-P65/P65 protein and so on, shouldbeincluded in the manuscript.
  2. RIP3 overexpression and Mitofilin knockdown decrease mitochondrial membrane potential and cell viability, increase cell death and activates cGAS/STING/NFKB pathway. Mitofilin overexpression suggest to be investigate to support the protective effect on RIP3 overexpression leading to cell death?On the other hand, mtDNA, p-65/p65 protein, inflammatory genes (IL-6, TNF-α and ICAM-1) and protein should be evaluatedin the cells of RIP3 overexpression and Mitofilin knockdown.
  3. The authors introducedthe statistical description in 2.16 Statistical Analysis “Comparisons were conducted using the Student’s t-test and two-way ANOVA with post-hoc Dunnett’s, Tukey’s, and Sidak’s corrections for multiple comparisons“, but the statistic test selected is not correct for the data. The data in these paper, the one-way ANOVA should be chosed and the post-hoc tests cannot use Dunnett’s, Tukey’s, and Sidak’s corrections at the same time. Please check and re-analyzethe data.
  4. The measure of mitochondrial membrane potential was assessed by the JC-1 in mitochondria and Mitotracker Red in HK2 cells. Why the authors did not use the same method JC-1 dye in HK2 cells. Besides, the levels of ROS and mitochondrial membrane potenntial should be evaluated in the cells of RIP3 overexpression and Mitofilin knockdown.
  5. The authors claimed the increase of RIP3 promote mitofilin degradation, but the mechanism underlying does not be investigated, which should be discussed.
  6. Zhou et.al found RIP3 positively regulated mitochondria-mediated apoptosis via FUNDC1 mitophage[2]; Wheeler et.alreported that RIP3 interacted to GLUD1, then activated GLUD1, increaseing Glu and Gln consumption as energy substrated[4]. These lead to an increase in energy metabolism and subsequent overproduction of the oxidative metabolism product, ROS. In themanuscript, the authors suggested RIP3/Mitofilin axis play important roles in kidney after I/R, so the effect of mitophage and GLUD1 in the RIP3/mitofilin pathway in kidney should be discussed.

Some minor concerns:

  1. The author described "the the left renal pedicle was exposed and clamped for 60 minutes with micro aneurysm clamps'' in the methods, but there were several ischemia time (30 min or 35 min) in the results. The authors must carefully check throughout the whole manuscript.
  2. In manuscript the authors presented the data using two graph forms (scatter plot and bar graph), the scatter plot may be betterfor all. Please carefully check the image data between Figure 1B and Figure 2B. Figure 1B legand description was wrong according to figure 1B. In my opinion, merging figure 1 and figure 2 into one would be better. Figure 4C and figure 10C, please add scatter plot and statistical result.
  3. Why the authors did use different number of samples during all the work in vivo? Please describe number of samples in every group in figure legend.
  4. In figure 6, IL-6, TNF-α and ICAM-1 protein levels in kidney should be evaluated using western or other methods.
  5. In result 3.2, the description "RIP3 plays an importnat role in the development of renal I/R injury observed in previous studies", please the authors add references.

  1. Chen, H.; Fang, Y.; Wu, J.; Chen, H.; Zou, Z.; Zhang, X.; Shao, J.; Xu, Y. RIPK3-MLKL-Mediated Necroinflammation Contributes to AKI Progression to CKD. Cell Death Dis2018, 9, 878, doi:10.1038/s41419-018-0936-8.
  2. Hao Zhou; Pingjun Zhu; Jun Guo; Nan Hu; Shuyi Wang; Dandan Li; Shunying Hu; Jun Ren; Feng Cao; Yundai Chen Ripk3 Induces Mitochondrial Apoptosis via Inhibition of FUNDC1 Mitophagy in Cardiac IR Injury. Redox Biol.2017, 13, 498–506, doi:10.1016/j.redox.2017.07.007.
  3. Liu, W.; Chen, B.; Wang, Y.; Meng, C.; Huang, H.; Huang, X.-R.; Qin, J.; Mulay, S.R.; Anders, H.-J.; Qiu, A.; et al. RGMb Protects against Acute Kidney Injury by Inhibiting Tubular Cell Necroptosis via an MLKL-Dependent Mechanism. Proc Natl Acad Sci U S A2018, 115, E1475–E1484, doi:10.1073/pnas.1716959115.
  4. D.-W. Zhang; J. Shao; J. Lin; N. Zhang; B.-J. Lu; S.-C. Lin; M.-Q. Dong; J. Han RIP3, an Energy Metabolism Regulator That Switches TNF-Induced Cell Death from Apoptosis to Necrosis. Science2009, doi:10.1126/science.1172308.

Author Response

Reviewer 2.

The  major concerns:

  1. RIP3 has been reported to interaction with receptor-interacting protein kinase 1(RIP1) and pseudokinase mixed lineage kinase domain-like protein (MLKL) to form the necrosome. The important roles of RIP3-MLKL-dependent necroptosis in acute kidney injury has been proved. The author declared RIP3 translocation in mitochondria to interact with Mitofilin and promote its loss as well as mitochondria dysfunction observed after AKI is undependently on the necrosome (RIP3/MLKL/RIP3). For these, the protective effect of MLKL inhibitor (Necrosulfonamide) on acute kidney injury is very important data. The manuscript only provided Figure-10C, which is not enough to support the results. Considered necrosulfonamide has not be reported in AKI, strongly suggest to provide the details of animal study, including dosage selection, route and times of administration et.al. The key data, including renal histopathology, mtDNA, inflammatory gene and protein(IL-6, TNF-α and ICAM-1), p-P65/P65 protein and so on, should be included in the manuscript.

We agree with the reviewer’s comments on the important role of mixed lineage kinase domain-like (MLKL) in the pro-inflammatory necroptotic cell death program. However, our results showed in (Fig. 10) indicate that MLKL does not play a major role in the regulation of the RIP/Mitofilin axis observed after renal I/R injury. Therefore, we believe that focusing on MKLK will not strengthen our manuscript. In addition, Figure 10 shows the impact of Necrosulfonamide (MLKL inhibitor) in the RIP3, Mitofilin, GAS and STING protein levels after renal I/R. We agree that it would be interesting to investigate the role of Necrosulfonamide in renal I/R injury in future studies Since MLKL does not mediate the RIP3/Mitofilin pathway, which is the target of the current manuscript.

  1. RIP3 overexpression and Mitofilin knockdown decrease mitochondrial membrane potential and cell viability, increase cell death and activates cGAS/STING/NFKB pathway. Mitofilin overexpression suggest to be investigate to support the protective effect on RIP3 overexpression leading to cell death?

We agree with the reviewer’s comment and have performed additional experiments by co-transfecting HK2 cells with RIP3-overexpressed and Mitofilin-overexpressed plasmids. We found that, the level of the cell viability was reduced when compared to RIP3-overexpressed alone (Figure S1C) suggesting that restoring the levels of Mitofilin can protect against cell death induced by RIP3 overexpression. Because this co-transfection did not fully rescue the cell viability level comparable to control vector suggesting that RIP3-induced cell death may activate alternative mechanisms. However, it is to emphasize that the in vivo effects of RIP3 translocation into mitochondria to cause cell death might be slightly different to the use of RIP3-overexpressed plasmid. Anyway, this data support the hypothesis that Mitofilin acts as a downstream in the RIP3 actions. Note that, there was no difference in the levels of cell viability in Mitofilin overexpressed plasmid versus control plasmid indicating that Mitofilin overexpression does not induce deleterious effects in cells. 

On the other hand, mtDNA, p-65/p65 protein, inflammatory genes (IL-6, TNF-α and ICAM-1) and protein should be evaluated in the cells of RIP3 overexpression and Mitofilin knockdown.

We usually do not measure inflammation markers in cell line. However, in a separate study, we determined the role of Mitofilin knockdown in the increase in inflammatory markers in the cardiac mice subjected to I/R injury in vivo.

The results presented in figure 2 below are only for the reviewer record. These unpublished are new data that indicate that Mitofilin heterozygote mouse mitochondria release more mtDNA (D-LOOP1) in the cytosol, and produce more inflammatory markers when compared to their littermate WTs.

  1. The authors introduced the statistical description in 2.16 Statistical Analysis “Comparisons were conducted using the Student’s t-test and two-way ANOVA with post-hoc Dunnett’s, Tukey’s, and Sidak’s corrections for multiple comparisons“, but the statistic test selected is not correct for the data. The data in these paper, the one-way ANOVA should be chosed and the post-hoc tests cannot use Dunnett’s, Tukey’s, and Sidak’s corrections at the same time. Please check and re-analyze the data.

We agree with the reviewer’s comment, and have edited the text accordingly. Data presented in bar graphs are expressed as means, and error bars are the standard errors of the mean (± SEM) for a minimum of three independent trials (n ≥ 3). Comparisons were conducted using the Student’s t-test and one-way ANOVA with post-hoc Dunnett’s or Tukey’s corrections for multiple comparisons, where appropriate, using Prism 8 (Graphpad Software). A difference of P < 0.05 was considered to be statistically significant.

  1. The measure of mitochondrial membrane potential was assessed by the JC-1 in mitochondria and Mitotracker Red in HK2 cells. Why the authors did not use the same method JC-1 dye in HK2 cells. Besides, the levels of ROS and mitochondrial membrane potential should be evaluated in the cells of RIP3 overexpression and Mitofilin knockdown.

We first measured mitochondrial membrane potential (MMP) with Mitotraker red in HK2 cells and decided thereafter to assess the MMP directly in mitochondria. Because, JC-1 would provide a better measure of the uncoupled mitochondria with the green fluorescence. We therefore, used JC-1 for the assessment of MMP in isolated mitochondria. We could also measure the MMP in HK2 cells; however, it would not add anything or provide a different outcome. We have already reported that Mitofilin knockdown increases the levels of ROS production and decreases mitochondrial membrane potential in rat cardiomyoblasts Am J Physiol Cell Physiol. 2018 Jul 1; 315(1): C28–C43 (Figure 6). Therefore, performing these experiments will not improve the present manuscript.

  1. The authors claimed the increase of RIP3 promote mitofilin degradation, but the mechanism underlying does not be investigated, which should be discussed.

We agree with the reviewer’s comment, and have added a small paragraph that discusses the potential mechanisms that could lead to Mitofilin degradation. In fact, we recently found an increase in Mitofilin ubiquitination in N27-A, and Human Dopamine Neuronal Primary cells treated with PD stressors, Dopamine or Rotenone. In addition, it has been reported that Mitofilin is highly subject to oxidation, which can result from the increase in oxidative stress. We postulate that RIP3 translation into mitochondria might increase Mitofilin ubiquitination or oxidation that result in its elimination.

  1. Zhou et.al found RIP3 positively regulated mitochondria-mediated apoptosis via FUNDC1 mitophage; Wheeler et.al reported that RIP3 interacted to GLUD1, then activated GLUD1, increasing Glu and Gln consumption as energy substrated. These lead to an increase in energy metabolism and subsequent overproduction of the oxidative metabolism product, ROS. In the manuscript, the authors suggested RIP3/Mitofilin axis play important roles in kidney after I/R, so the effect of mitophage and GLUD1 in the RIP3/Mitofilin pathway in kidney should be discussed.

We agree with the reviewer’s comment, and have added a small paragraph that discusses the RIP3/Mitofilin axis and energy metabolism.

 Some minor concerns:

  1. The author described "the left renal pedicle was exposed and clamped for 60 minutes with micro aneurysm clamps'' in the methods, but there were several ischemia time (30 min or 35 min) in the results. The authors must carefully check throughout the whole manuscript.

We agree with the reviewer’s comment, and have edited the text accordingly. In our protocol, the left renal pedicles were exposed and clamped for 60 minutes with micro aneurysm clamps.

  1. In manuscript the authors presented the data using two graph forms (scatter plot and bar graph), the scatter plot may be better for all. Please carefully check the image data between Figure 1B and Figure 2B. Figure 1B legand description was wrong according to figure 1B. In my opinion, merging figure 1 and figure 2 into one would be better. Figure 4C and figure 10C, please add scatter plot and statistical result.

We agree with the reviewer’s comment, and have edited the text accordingly. However, merging figure 1 and figure 2 into one would change all the structure of the manuscript. Therefore, we decided to keep the two figures separate.

  1. Why the authors did use different number of samples during all the work in vivo? Please describe number of samples in every group in figure legend.

We agree with the reviewer’s comment. For all the in vivo studies, we have performed new experiments, which resulted in a slightly different number of samples. In general, we have used a minimum of three separate experiments/group for biochemical analysis.

  1. In figure 6, IL-6, TNF-α and ICAM-1 protein levels in kidney should be evaluated using western or other methods.

We partially agree with the reviewer’s comment. If Western blot analyses the protein levels, our hypothesis is that increase in mtDNA release into the cytosol in response to Mitofilin degradation leads to increased transcription of pro-inflammatory markers including IL-6, TNF-α and ICAM-1 in the nucleus. Therefore, quantifying the mRNA by qRT-PCR is valuable approach. We do not see the reason the protein levels would be different to mRNA levels. Therefore, Western blot analysis would only confirm our data using qRT-PCR.

  1. In result 3.2, the description "RIP3 plays an important role in the development of renal I/R injury observed in previous studies", please the authors add references.

We agree with the reviewer’s comment, and have edited the text accordingly.

Round 2

Reviewer 2 Report

All comments were well addressed.